# Chagas Disease Megaesophagus Patients Carrying Variant *MRPS18B* P260A Display Nitro-Oxidative Stress and Mitochondrial Dysfunction in Response to IFN-γ Stimulus

**DOI:** 10.3390/biomedicines10092215

**Published:** 2022-09-07

**Authors:** Karla Deysiree Alcântara Silva, João Paulo Silva Nunes, Pauline Andrieux, Pauline Brochet, Rafael Ribeiro Almeida, Andréia Cristina Kazue Kuramoto Takara, Natalia Bueno Pereira, Laurent Abel, Aurelie Cobat, Ricardo Costa Fernandes Zaniratto, Débora Levy, Sergio Paulo Bydlowski, Ivan Cecconello, Francisco Carlos Bernal da Costa Seguro, Jorge Kalil, Christophe Chevillard, Edecio Cunha-Neto

**Affiliations:** 1Laboratory of Immunology, Heart Institute (Incor) Hospital das Clínicas da Faculdade de Medicina da Universidade de São Paulo, São Paulo 05403-900, Brazil; 2Division of Clinical Immunology and Allergy, Faculdade de Medicina da Universidade de São Paulo, São Paulo 05403-000, Brazil; 3Institute for Investigation in Immunology/INCT, São Paulo 05403-900, Brazil; 4Institut MarMaRa, INSERM, UMR_1090, Aix Marseille Université, TAGC Theories and Approaches of Genomic Complexity, 13288 Marseille, France; 5Laboratory of Human Genetics of Infectious Diseases, Necker Branch, INSERM U1163, Necker Hospital for Sick Children, 75015 Paris, France; 6Imagine Institute, University of Paris, 75015 Paris, France; 7St. Giles Laboratory of Human Genetics of Infectious Diseases, Rockefeller Branch, The Rockefeller University, New York, NY 10065, USA; 8Hospital das Clínicas da Faculdade de Medicina da Universidade de São Paulo, São Paulo 05403-900, Brazil

**Keywords:** chagasic megaesophagus, mitochondria, mitochondrial mutation, mitochondrial dysfunction, interferon-gamma, MRPS18B, Chagas disease, *Trypanosoma cruzi*

## Abstract

Chagas disease (CD), caused by the protozoan parasite *Trypanosoma cruzi*, affects 8 million people, and around 1/3 develop chronic cardiac (CCC) or digestive disease (megaesophagus/megacolon), while the majority remain asymptomatic, in the indeterminate form of Chagas disease (ASY). Most CCC cases in families with multiple Chagas disease patients carry damaging mutations in mitochondrial genes. We searched for exonic mutations associated to chagasic megaesophagus (CME) in genes essential to mitochondrial processes. We performed whole exome sequencing of 13 CME and 45 ASY patients. We found the damaging variant *MRPS18B* 688C > G P230A, in five out of the 13 CME patients (one of them being homozygous; 38.4%), while the variant appeared in one out of 45 ASY patients (2.2%). We analyzed the interferon (IFN)-γ-induced nitro-oxidative stress and mitochondrial function of EBV-transformed lymphoblastoid cell lines. We found the CME carriers of the mutation displayed increased levels of nitrite and nitrated proteins; in addition, the homozygous (G/G) CME patient also showed increased mitochondrial superoxide and reduced levels of ATP production. The results suggest that pathogenic mitochondrial mutations may contribute to cytokine-induced nitro-oxidative stress and mitochondrial dysfunction. We hypothesize that, in mutation carriers, IFN-γ produced in the esophageal myenteric plexus might cause nitro-oxidative stress and mitochondrial dysfunction in neurons, contributing to megaesophagus.

## 1. Introduction

Chagas disease is caused by the flagellate protozoan *Trypanosoma cruzi* and the transmission occurs through the bite of triatomine vectors [1]. Patients infected with *T. cruzi* develop acute and chronic forms of the disease. The acute phase is characterized by high parasitemia and increased production of the pro-inflammatory cytokine interferon-gamma (IFN-γ) [2]. In the chronic phase, a proportion of the infected patients remain asymptomatic, without cardiac or digestive damage—in the so-called indeterminate form of Chagas disease (ASY)—while 30% of patients end up developing serious damage to the heart —chronic Chagas cardiomyopathy (CCC)—associated with heart failure, cardiac electrical conduction disorders and cardiac arrhythmias [3]. About 10% of patients develop digestive tract motility disorders, most often dilatations in the esophagus and colon (megaesophagus and megacolon, respectively). The digestive forms of Chagas disease cause a considerable decrease in quality of life, often requiring surgical intervention [4,5]. In both cases, there is degeneration of the myenteric plexuses and destruction of neurons that control the motility of that segment of the digestive tract [6].

The differential susceptibility of 30% of those infected by *T. cruzi* to evolution to CCC and 10% to digestive disease, while most patients remain in the indeterminate form, suggests the participation of host genetics. This possibility was reinforced by the finding of family aggregation of CCC cases [7]. More than 150 case–control studies on CCC have been published, studying common genetic polymorphisms in over 70 candidate genes, mainly related to the immune response [8]. Only two studies, however, evaluated genetic polymorphisms associated with digestive forms of Chagas disease, ficolin-2 [9] and HLA-G [10].

The scarcity of data highlights the importance of studying the genetic aspects of the digestive form of Chagas disease. Recent results from our group pioneered the use of the whole exome sequencing (WES) methodology to investigate rare genetic variants associated with CCC in families with multiple cases of Chagas disease [11]. We observed that, in each family, patients with CCC shared rare heterozygous, non-synonymous and pathogenic variants among them. Such genetic variants were absent from family members with ASY, without cardiac involvement. Of the 25 CCC specific pathogenic variants observed, 11 occurred in mitochondrial genes and 14 in inflammatory genes; CCC-specific mitochondrial gene variants were found in five out of the six families studied. 

Mitochondria play a central role in energy supply, metabolic regulation and cell death in response to harmful stimuli [12]. Mitochondriopathies, caused by homozygous pathogenic variants in genes encoded in the nucleus and mitochondrial DNA, are the most common monogenic syndromes. They severely impair energy metabolism and cause mitochondrial dysfunction and functional deterioration in tissues with high metabolic demand, such as cardiac, skeletal muscle, nervous tissue and liver [13,14]. It is interesting to notice that 30–40% of cases of mitochondrial diseases progress to cardiomyopathy, cardiac conduction disorders and severe arrhythmias [15,16] and 15% for digestive motility disorders, including megacolon and megaesophagus with denervation of the myenteric nervous plexuses, and autonomic nervous system disorders [17,18]. The similarity in the nature and prevalence of cardiac, digestive and autonomic nervous system involvement in mitochondriopathies and those observed in the evolution of Chagas disease suggests that heterozygous pathogenic genetic variants, which cause a partial reduction in mitochondrial function, may play a role in the differential evolution of Chagas disease, both for CCC and for the digestive forms.

Studies show that inflammatory cytokines such as IFN-γ and TNF-α cause mitochondrial damage in several cell types [19,20]. IFN-γ and TNF-α, which are abundantly produced in Chagas disease in response to the perennial stimulus of *T. cruzi* [2,21,22], cause a reduction in oxidative metabolism, in the expression of energy metabolism enzymes, in the activity of lipid beta-oxidation and in the production of mitochondrial ATP, and reduction in the action of mitochondrial membrane potential (ΔΨm) via the NF-κB-dependent signaling pathway [3,23,24,25,26,27,28,29]. IFN-γ induces superoxide generation by increasing the expression of NADPH oxidases [30] and induces NO generation by upregulating nitric oxide synthase (NOS2) [31]. Superoxide reacts with NO to produce the highly reactive nitrogen species peroxynitrite, which can destroy *T. cruzi* but also can cause host cell and mitochondrial damage. We hypothesized that pathogenic mutations in genes essential to mitochondrial processes were associated with the digestive form of Chagas disease. To evaluate our hypothesis, we performed whole exome sequencing in carriers of CME and the indeterminate form of Chagas disease. To assess whether cells from patients carrying these mutations were more likely to suffer cytokine-induced nitro-oxidative stress and mitochondrial dysfunction, we also evaluated the effect of IFN-γ in the Epstein–Barr virus-immortalized B lymphoblastoid cells (EBV-LCL) obtained from patients with CME carrying or not carrying mutations in mitochondrial genes. These cell lines have been used to study mitochondrial dysfunction in genetic mitochondriopathies [32].

## 2. Methods

### 2.1. Patients and Blood Collection

Patients with Chagas disease megaesophagus (CME) from the Division of Digestive Tract Surgery of Hospital das Clínicas da Faculdade de Medicina da Universidade de São Paulo (HC-FMUSP) were invited to participate in the study. An informed consent form was applied if the patient wanted to participate in the study. The study was approved by ethical committee (by the number 35658720.2.0000.0068) and all patients selected for this study had at least two serological tests for Chagas disease (ELISA and/or indirect immunofluorescence—IFI) performed by the clinical analysis laboratory of HC-FMUSP. 

The control group was made up of patients with the indeterminate form of Chagas disease (ASY) recruited at the Chagas Disease Outpatient Clinic at the General Cardiomyopathies Center of the Heart Institute (InCor), Hospital das Clínicas da Faculdade de Medicina da Universidade de São Paulo. 

### 2.2. Whole Exome Sequencing (WES) of Patients with CME and ASY

DNA was extracted from whole blood of CD patients with the QIAamp DNA Blood Maxi Kit (Qiagen, Redwood City, CA, USA). The exome of 13 CME patients and 45 ASY patients was sequenced by Genewiz (Paris, France). Low-quality adapter sequences and nucleotides were removed using Trimmomatic v.0.36. The contigs were aligned to the reference genome Homo sapiens GRCh37 using the Edico Genome Dragen’s pipeline. Variant call format (VCF) files were used for the annotations and prioritization of mutations in the VarAFT software (version 2.06, Windows 10 platform) (https://varaft.eu; 29 April 2019) which generates a file for each individual with a list of all variants detected from several databases such as 1000 Genomes and ExAC. Pathogenic variants were performed using the following criteria based on the markers that were considered significant: maintenance of variants located in exon and splicing zones, elimination of synonymous variants, four different databases (SIFT, UMD and two PolyPhen) and the selected mutation considered pathogenic in at least 3 of them. The MRPS18B gene of CMECME and ASY patients was amplified by real-time PCR using the following primers: F 5′-TACCTCTCATGGGGCTGTGAGT-3′ and R 5′-TTCCGCTGGTGTTCTAGGGGG-3′. The resulting amplicon was subjected to Sanger sequencing (Genewiz, Leipzig, Germany). EBV-derived lymphoblastoid cell lines (EBV-LCL) from four patients carrying MRPS18B P260A (one homozygous and the other 3 heterozygous) and three wild type CME patients were generated for downstream analyses.

### 2.3. Variants Selection

An average of 140,677 annotated variants were detected in each CME patient whereas an average 136,177 annotated variants were detected in each asymptomatic subject. In total, 456,344 variants were detected in at least one of the 13 CME patients whereas 646,398 variants were detected in at least one of the 37 ASY subjects.

According to our hypothesis, we focused our analysis only on genes that are involved in mitochondrial functions. The mitochondrial gene list was a combination of all genes contained in the Mitochondrion Gene Ontology term and Mitocarta 2.0 (see Appendix A). This gene list contains 1531 genes which can be included either in the nuclear genome or in the specific mitochondrial circular genome. Among the CME variants, a small fraction (6.6% = 30,184/456,344) of the variants were included in our candidate genes.

Among these candidate variants, we selected the variants located in coding (exonic or splicing) regions and that were non-synonymous. For each of these variants, we looked for pathogenicity data available in four databases (Shift, PolyPhen2_HDIV_pred, PolyPhen2_HVAR_pred and UMD predictor). We selected the variants described as pathogenic (or damaging) in at least three of these databases. So, the number of candidate variants dropped to 217 pathogenic non-synonymous variants located in exonic or splicing regions of one of our candidate genes.

In 2014, the CADD score was introduced as a comprehensive tool that aims to take the results of many known prediction tools into account. Here, we focus our analysis on variants with a CADD score above 20. A scaled CADD score of 20 means that a variant is amongst the top 1% of deleterious variants in the human genome. In our statistical analysis, we compared the number of mutated chromosomes in each group. The cutoff value for statistic tests was fixed at 0.005.

### 2.4. Isolation of Peripheral Blood Mononuclear Cells 

Peripheral blood mononuclear cells (PBMCs) were purified from 20 mL of blood collected in tubes treated with heparin from CME patients. Blood was diluted in 0.9% saline solution (1:1 dilution) and carefully superimposed on Ficoll-Hypaque (Amersham Biosciences, Uppsala, Sweden) and submitted to centrifugation at 800× *g* (without braking and without acceleration) for 30 min for the separation of PBMCs from red blood cells and plasma. The PBMCs were collected from the interphase with the aid of a Pasteur pipette and washed 3 times in 0.9% saline at 300× *g* 8 min. Cells were counted and subjected to immortalization using Epstein–Barr virus. 

### 2.5. Production of a Supernatant Rich in Epstein–Barr Virus Particles

To enrich the supernatant with Epstein–Barr virus (EBV) particles, approximately 2 × 10^6^ monkey B95.8 cells purchased from Banco de Células do Rio de Janeiro (BCRJ) were grown in specific medium R15 (RPMI 1640 (Gibco^®^, New York, NY, USA) supplemented with 15% inactivated FBS (Life Technologies^TM^, New York, NY, USA), 10 U/mL penicillin, 100 µg/mL streptomycin and 0.25 µg/mL amphotericin B (Thermo Fisher Scientific, Waltham, MA, USA)) at a density of 0.2 × 10^6^ cells per ml in R15 medium in a flask with 25 cm² of growth surface (T25). The cells were kept in an incubator at 37 °C with 5% CO_2_ for seven days and the medium was transferred to a new 75 cm^2^ flask pre-filled with 10 mL of R15 medium. Once a week, 10 mL of R15 was added until reaching 50 mL. Then, the flask was completely closed and kept in the incubator at 37 °C for 14 days. The medium was then collected and filtered through a 0.22 µm filter to eliminate debris. Aliquots of 1 mL of the supernatant were frozen and stored at −80 °C. 

### 2.6. Establishment of EBV-LCL from Patients 

The PBMCs isolated from all patients were infected with the EBV for transformation of B cells into EBV-LCL. EBV-LCL are continuous sources of cells for genetic and functional studies [33] and have been widely used for the study of mitochondrial function in mitochondrial diseases [32,34]. Transformation was carried out as follows: 1.0 × 10^6^ viable PBMCs were incubated in 400 µL of R20 medium (RPMI 1640 (Gibco^®^, New York, NY, USA)) supplemented with 20% inactivated FBS (Life Technologies^TM^, New York, NY, USA), 10 U/mL of penicillin, 100 µg/mL of streptomycin and 0.25 µg/mL of amphotericin B (Thermo Fisher Scientific, Waltham, MA, USA), with 200 µL of the EBV-enriched supernatant and 2.4 µL of cyclosporin A 1.0 mg/mL (Sigma-Aldrich, Saint Louis, MI, USA) in a 1.9 cm^2^ surface plate (24-well plate). The cells were incubated at 37 °C with 5% CO_2_ for seven days without disturbance. 

On the 7th day, 250 µL of R20 medium was added to each well and incubated for another seven days in an incubator at 37 °C with 5% CO_2_ without disturbance. On the 14th day, the entire content of the three wells was transferred to a 12.5 cm^2^ flask (T12.5) pre-filled with 1 mL of R20 and the cells were incubated at 37 °C with 5% CO_2_. Every two or three days, 200–500 µL of R20 was gradually added depending on the growth rate of the cells.

When the culture volume reached 4 mL, cells were centrifuged at 100× *g* for 10 min at room temperature and then the supernatant was discarded. Cells were suspended at a density of 0.4 × 10^6^ viable cells/mL in R15 medium and left to grow until 20 × 10^6^ cells. Then, cells were cryopreserved with 90% FBS (Gibco^®^, New York, NY, USA) and 10% DMSO (Sigma-Aldrich, Saint Louis, MI, USA) in liquid nitrogen for further analyses.

### 2.7. Evaluation of mtROS in LCLs

Twenty-four hours before the experiment, the EBV-LCL cells from each patient were centrifuged at 100× *g* for 10 min and the counting was carried out with a Neubauer chamber and trypan blue 0.4% (Invitrogen, New York, NY, USA). The cells were kept at a concentration of 0.4 × 10^6^ viable cells per ml of R15.

On the day of the experiment, the cells were centrifuged at 100× *g* for 10 min, counted and seeded at a concentration of 15 × 10^4^/mL in 150 µL of R15 containing or not containing interferon-γ (IFN-γ, 25 ng/mL) in sterile 96-well U-bottom plates. After stimulation, cells were transferred to V-bottom 96-well plates, centrifuged at 300× *g* for 5 min and suspended in Hanks saline buffer with Ca^2+^ and Mg^2+^ pH 7.4 (HBSS++) containing 5 µM of MitoSOX™ Red (Thermo Fisher Scientific, Waltham, MA, USA), specific for mitochondrial superoxide. The cells were then incubated in the dark at 37 °C for 10 min.

Afterwards, the plate was centrifuged at 300× *g* for 5 min and the cells were resuspended in 150 µL of HBSS++ containing 3.5 µM of Hoechst 33342 (ThermoFisher Scientific, Waltham, MA, USA) and 1 drop per ml of NucGreen (ReadyProbres, Thermo Fisher Scientific, Waltham, MA, USA). The stained cells were transferred to 96-well black plates (Corning 3603) at 3 × 10^4^ per well in 100 µL of HBSS++ and centrifuged at 300× *g* for 2 min without acceleration and braking. Images of five fields per well were captured in a high-content screening system (HCS, ImageXpress, Molecular Devices, Sunnyvale, CA, USA) at 200× magnification using Hoechst 33342 (nucleus), FITC (dead cell nucleus) and Texas red (mtROS) filters.

The MitoSOX signal was measured using the software Columbus. First, we identified the total number of cells by using the nucleus probe Hoechst 33342. Then, dead cells were excluded by using a second probe that stains the nucleus green in dead cells. By excluding dead cells, we measured MitoSOX fluorescence intensity inside live cells at approximately 580 nm (Texas red). The contrast intensity (c) is automatically calculated by Columbus with the following function of the mean intensity in the region (a), and the mean intensity in the neighborhood of the region (b): c = (a − b)/(a + b). Approximately 9000–10,000 cells were measured per patient.

### 2.8. Sample Preparation of EBV-LCL Stimulated with IFN-γ

For concurrent quantification of nitrite, ATP and nitrated proteins, 50 × 10^4^ cells from each patient were seeded in 2.5 mL of R15 without phenol red in 6-well plates. Cells were treated with IFN-γ (25 ng/mL) for 48 h. After treatment, cells were collected, divided into two 1.5 mL microtubes and centrifuged at 800× *g* for 5 min. Conditioned supernatant was collected in 1.5 mL microtubes and stored in a freezer at −80 °C until use for nitrite quantification.

The pellets formed in the two 1.5 mL microtubes were processed in different ways on ice: (1) the first cell pellet was lysed in 200 µL of TE buffer (100 mM Tris and 4 EDTA, pH = 7.5). Afterwards, the lysate was heated at 95 °C for 7 min and centrifuged at 20,800× *g* for 3 min and the supernatant was collected and then stored in a freezer at −80 °C until use for ATP quantification; (2) the second cell pellet was lysed in 200 µL of lysis buffer (10 mM HEPES, 0.32 M sucrose, 0.1 mM EDTA, 1.0 mM dithiothreitol (DTT), 5.25 μL leupeptin, 0.3 μM aprotinin and 125 μg/mL PMSF at pH 7.4) with the aid of a cell disruptor (VirSonic 100 (Virtis, Warminster, PA, USA)) for 3 cycles of 10 s with 15 s intervals on ice and centrifuged at 10,000× *g* for 30 min at 4 °C and the supernatant was collected for quantification of nitrated proteins.

For both lysates, protein concentration was determined by the Bradford method (Bio-Rad, Hercules, CA, USA) and samples were stored at −80 °C until use.

### 2.9. Quantification of Nitrite, ATP and Nitrated Protein

The quantification of nitrite in the supernatant was performed using the Griess Reagent Kit (Thermo Fisher Scientific, Waltham, MA, USA) and ATP production was quantified using the ATP Determination Kit (Thermo Fisher Scientific, Waltham, MA, USA) as per the manufacturer’s instructions.

For the quantification of nitrated proteins, 5 µg of the protein was vacuum added to a nitrocellulose membrane 0.45 µm with the aid of a HYBRID DOT MANIFOLD dot blot apparatus (Bethesda Research Laboratories, Bethesda, MD, USA). The membrane was air dried in an Eppendorf ThermoMixer^®^ thermoblock (Thermo Fisher Scientific, Waltham, MA, USA) at 60 °C for 1 h and then blocked with 5% Blotting-Grade Blocker (Bio-Rad, Hercules, CA, USA) + TBS-Tween (NaCl 120 mM, Tris 20 mM, Tween20 0.05%) for 2 h at room temperature (RT), with agitation. After, membrane was washed 3 times for 5 min under agitation with TBS-T. The membrane was then incubated with 1:1000 of the primary antibody 3 free nitrotyrosine (3-NT) in BSA 0.3% + TBS-T overnight at 4 °C under agitation. After, membrane was washed three times and incubated with 1:10,000 of secondary antibody IRDye 680RD Donkey anti-mouse 700 nm (LI-COR, Lincoln, NE, USA) in 0.3% BSA + TBS-T for 2 h at RT, under agitation and protected from light. After, membrane was washed three times again and fluorescence was acquired using the Odyssey^®^ Infrared Imaging System scanner (LI-COR, Lincoln, NE, USA) in the 700 nm channel. The normalization was carried out using Ponceau staining in the ImageQuant LAS-400 image reader (GE Healthcare, Chicago, IL, USA).

### 2.10. Statistical Analysis 

Data and statistical analysis were performed using the program GraphPad Prism 8 (Windows 10). A Mann–Whitney test of the mean row statistics in the quantification of mtROS and a Mann–Whitney test for the quantification of nitrite, ATP and nitrate proteins for each patient were carried out. After, a second analysis was carried out per group, C/C (wild type), C/G (heterozygous) and G/G (homozygous), and the mean of each analysis was used. For this second analysis, we used an ANOVA two-way test. For exome analysis, two statistical tests were used: Fisher test and chi-squared. Values of *p* < 0.05 were considered significant.

## 3. Results

### 3.1. Exome Sequencing Reactions for CME Patients and ASY Subjects

We performed whole exome sequencing of 13 CME patients and 37 ASY subjects to identify if CME patients display an accumulation of pathogenic polymorphisms in comparison with ASY. The workflow is described in Figure 1.

Two variants were more frequent in patients with Chagas disease megaesophagus (*MRPS18B* rs34315095C/G and *FAM185A* rs201667800T/G) (Table 1, Table 2 and Table 3). The mitochondrial ribosomal protein S18B (MRPS18B) gene encodes a 28S subunit protein that belongs to the small subunit of the mitochondrial ribosome. We have little functional information on the FAM185A gene. A FAM185A variant was found in a study (whole mitochondrial genome sequencing, in conjunction with high-throughput genotyping arrays) carried out to discover genetic variants associated with exercise responses. The link with mitochondria is in fact not obvious [35]. So, we decided to focus on the MRPS18B gene, as many homozygous mitochondrial ribosome mutations are associated with genetic mitochondriopathies.

### 3.2. Pathogenic Variant of MRPS18B Is Accumulated in CME Patients 

The *MRPS18B* variant (688C > G, P230A) was present in 38% of CME patients (five of the 13 CME patients) and 3% of ASY patients (1 of 37) (Table 2). One of the five CME patients carrying the mutation was homozygous (G/G) while the other four CME carriers and the ASY carrier were heterozygous (C/G). So, the G allele frequency in CME is 23.1%, whereas this allelic frequency reaches 1.4% in ASY subjects.

The G variant rs34315095 has a frequency of 4.6% in the 1000 Genomes database. Based on the ALFA allele frequency project, in reference populations the alternative allele (G) frequencies were 1.8% (European reference population), 5.5% (African reference population); 5.2% (African American reference population) and 3.8% (Asian reference population). Recently, the ABraOM repository was built, containing genomic variants obtained with whole exome and whole genome sequencing from SABE, a census-based sample of elderly individuals from São Paulo, Brazil [36]. In both databases (exome sequencing database: 609 individuals and whole genome sequencing: 1171 individuals), the G frequency reaches 4.7%.

The confirmatory sequencing was performed by Sanger sequencing (Genewiz). The DNA was amplified by real-time PCR using specific primers for *MRPS18B*. The resulting amplicon was subjected to Sanger sequencing and results confirmed the presence of the polymorphism 680C > G of the MRPS18B gene (Figure 2 shows representative chromatograms). 

Chromatograms of all CME patients are depicted in Appendix A. 

### 3.3. IFN-γ Causes Nitro-Oxidative Stress and Mitochondrial Damage in Patients Carrying MRPS18B P260A Variation

In order to investigate if patients carrying the variant 688C > G *MRPS18B* are more susceptible to IFN-γ’s damaging effects compared to patients carrying the wild type genotype (C/C), we conducted functional assays in EBV-LCL derived from CME heterozygous (C/G) and homozygous (G/G) patients. We quantified mtROS, nitrite, nitrate proteins and synthesis of ATP in non-stimulated and IFN-γ-stimulated cells (25 ng/mL).

We found that C/G (CME06, CME08 and CME12) and G/G (CME07) patients are more susceptible to IFN-γ’s damaging effects compared to C/C (CME01, CME02 and CME03) patients. We identified that C/G and G/G patients had increased nitrite production (31% and 38%, respectively) as compared with the C/C group (Figure 3). Similarly, the C/G patients had a 67% increase and the G/G CME7 patients a 70% increase in nitrated proteins when compared with the C/C group (Figure 3B).

We next measured ATP production by luciferase-based assay and we found that IFN-γ reduced the ATP production 3% in C/G patients compared to C/C patients, while the CME07 G/G showed a reduction of 54% (Figure 3C) as compared to baseline conditions.

We evaluated mtROS production using MitoSOX™ Red, which binds preferentially to the superoxide radical (O₂^−^). Results show there is no significant increase in C/G vs. C/C patients, but homozygous patient CME7 (G/G) showed a 12% increase when compared with the other patients (Figure 3D and Figure 4A). The p value is 0.0022 with a Mann–Whitney test of six independent measurements of IFN-gamma-stimulated cells compared to unstimulated cells. The cell viability assay indicated no cytotoxic effect of IFN-γ at the concentration of 25 ng/mL (Figure 5). 

## 4. Discussion

This study demonstrated for the first time the relevance of pathogenic variants in patients with Chagas disease megaesophagus. We demonstrate that CME patients have a higher frequency of pathogenic mutations compared to FI patients, such as the mitochondrial mutation MRPS18B 688C > G which was identified in 38% of CME patients and 2% of FI patients. The MRPS18B variant (688C > G, P230A) was characterized by a Sift score of 0.01 (damaging), a PolyPhen score of 0.996 (damaging) and a CADD score of 28. The ENST00000259873: exon7: 688C base is well conserved in the evolution (score 8/9). The proline 230 is located in an external alpha helix.

IFN-γ treatment in EBV-LCL cell lines carrying MRPS18B 688C > G was associated with increased nitro-oxidative stress compared to patients with megaesophagus without the mutation and the homozygous patient displayed higher damaging effects as shown by decreased ATP production and superoxide production. 

In this work, we focused on evaluating the impact of variants of MRPS18B, because recent studies link mitochondrial dysfunction and poor prognosis of Chagas disease. The MRPS18 proteins are encoded by nuclear genes and are involved in protein synthesis within mitochondria. The homologous proteins are zinc-binding proteins that make up the 28S small subunit (MRPS18B and C) and the 39S large subunit (MRPS18A) of the 55S mammalian mitoribosomes [37,38,39,40]. Mitochondria have their own translation system for the synthesis of thirteen proteins that build the oxidative phosphorylation complex (OXPHOS). Mutations in mitoribosomal proteins are frequent causes of metabolic and energy disturbances due to deficiencies in mitochondrial protein synthesis [41]. Mutations in the MRPS genes have been shown to have deleterious effects, especially in disorders of the mitochondrial respiratory chain. Mitochondrial disease-causing mutations were found in thirteen small ribosomal subunit mitoribosomal proteins (MRPS1, MRPS2, MRPS6, MRPS7, MRPS9, MRPS11, MRPS14, MRPS16; MRPS22, MRPS23, MRPS25, MRPS34, MRPS39) and four large ribosomal mitochondrial proteins (MRPL3, MRPL12, MRPL24, MRPL44) [41]. Several of them are associated with cardiomyopathy, as was also observed in a study by our group where the presence of a different mutation in the MRPS18B gene was identified in CCC patients [11,41]. Some proteins in this group play extraribosomal roles, such as participating in the regulation of apoptosis and a series of cell signaling cascades [38]. Mutations in the mitochondrial genes that affect mitochondrial translation, such as in mitochondrial tRNAs, have been associated with Leigh syndrome and mitochondrial neurogastrointestinal encephalopathy (MNGIE), two mitochondrial diseases with loss of myoteric plexus neurons and dysphagia [13,18,41,42].

Our results corroborate the role of mitochondrial gene variants in susceptibility to symptomatic forms of Chagas disease, as observed for the first time in CCC patients [11]. However, most mitochondrial diseases affect young children and newborns and often involve multiple organ systems. Interestingly, patient CME07, who carries the homozygous variant, is a 60-year-old patient with digestive and cardiac symptoms (heart failure, New York Heart Association (NYHA) functional class II). The fact that the mitochondrial phenotype in vitro only emerged after cytokine stimulation and was only clinically expressed in an adult is suggestive of a two-hit phenomenon, where the mutation increases the cells’ susceptibility to nitro-oxidative stress induced by IFN-γ and mitochondrial dysfunction. Thus, the two-hit mitochondrial damage would be located only in inflammatory sites in patients with Chagas disease; for example, in myenteric plexuses in patients with Chagas disease with megaesophagus [6] and in the heart in the case of CCC [43]. The MRPS18B gene is one of the variant genes identified only in the CCC siblings of one Chagas multicase family, mentioned above [11].

We postulate that CME patients display pathology because this mutation causes mitochondrial ribosome dysfunction, which affects protein synthesis essential for adequate oxidative phosphorylation and mitochondrial homeostasis. Defective oxidative phosphorylation may first lead to increases in ROS which may in turn activate NF-kB and NOS2, inducing increased levels of NO which under high ROS conditions can generate the powerful oxidant peroxynitrite, causing oxidative damage to the cell and mitochondria.

Chronic immune response is also a key factor associated with worse progression of patients with Chagas disease [44]. The pro-inflammatory cytokines IFN-γ and TNF-α are increased in patients with CCC and digestive disease compared to patients with dilated cardiomyopathy [21,45] and these stimuli are reported to be mitochondrial dysfunction inducers by causing damage to cardiac tissue [3,24,27] and, in in vitro cardiomyocytes [46], promoting an increase in the production of reactive oxygen species (ROS) and nitrogen (RNS) [45,47,48,49]. Considering that the trigger shared by patients with megaesophagus and Chagas cardiomyopathy is infection with *T. cruzi*, our hypothesis is that these CME patients may also have mitochondrial/energy dysfunction associated with pathology plus mutation in the mitochondrial gene.

We identified that CME patients when exposed to IFN-γ respond differently in the production of cytoplasmic mtROS. Although we did not find differences in the production of mtROS in the group of patients heterozygous for MRPS18B, we found that the CME07 MRPS18B G/G had a significant increase of 12% in mtROS compared to all groups, showing that the homozygous mutation may contribute to a worsening in mitochondrial metabolism that culminates in increased production of mtROS.

A study carried out to evaluate the role of nitric oxide in neuronal destruction in C57BL/6 control mice and knockout for NOS2 and IFN during *T. cruzi* infection demonstrated that the myoenteric denervation observed in the acute phase of protozoan infection is due to the production of nitric oxide (NO) and IFN-γ, as it was observed that control mice produced more NO when compared to the two knockout groups [50]. Our study showed that the EBV-LCL of CME patients carrying the genetic variant in homozygosis and heterozygosis showed an increase in the production of nitrite (product of endogenous NO metabolism) after treatment with IFN-γ, thus showing that the cytokine has an essential role in the increase in nitrite production and, consequently, generating a possible mitochondrial dysfunction and the increase in agents that in large quantities become toxic for the functioning of the mitochondria.

In addition to the increase in nitrite, patients with MRPS18B 688C > G P260A increased the amount of nitrated proteins after stimulation with IFN-γ compared to non-carriers. The identification of nitrated proteins is a consequence of the increase in peroxynitrite (ONOO^−^), a product originated by the combination of superoxide and NO or by the combination of hydrogen peroxide (H_2_O_2_) with nitrite [51]. Excessive production of ONOO^−^ can lead to tissue damage and mitochondrial dysfunction [51,52]. A study carried out in hepatoma and fibrosarcoma strains also showed that after treatment with IFN-γ there was an increase in peroxynitrite [51].

In previous studies, our group demonstrated that patients with CCC display a reduction in mitochondrial enzymes involved in ATP production [53]. Our investigations showed that this decrease may also be present in CME patients in sites of increased IFN-γ production, where we observed a decrease in ATP synthesis in the EBV-LCL of patients with mitochondrial mutation after treatment with IFN-γ. In fact, the homozygous patient (CME07 G/G) followed this pattern, where IFN-γ stimulation increased mtROS, oxide nitric and tyrosine nitration accompanied by reduced ATP production. The heterozygous group (C/G) increased nitrite and nitrotyrosine in an mtROS-independent way. This could be explained by direct activation of NOS2 by IFN-γ itself or fast catalytic conversion of leaked mtROS into H_2_O_2_ by antioxidant enzymes such as superoxide dismutase [54]. The increased protein nitration observed in our model could also be explained by combinatory damaging effects between enhanced NF-kB/NOS2 activation by IFN-γ. The employment of refined methodologies for mtROS detection, such as LC-MS and a “probe-free” electron paramagnetic resonance (EPR)/spin-trapping technique [55], could give us a better clue on mtROS production in these cells.

Our study had limitations. We focused our study on the genes which are essential for mitochondria. We cannot exclude the involvement of additional variants. The small number of analyzed CME patients may mask the identification of additional mitochondrial mutations and the reduced number of LCL used for functional assays presents only suggestive and not definitive results. However, the differences identified in the frequency of genetic variants between the CME (34 variants) and FI groups were robust, and a cellular/molecular phenotype was associated with the variant. To estimate if this pathway could indeed be influencing the EBV-LCL responses of patients, additional tests need to be carried out, such as silencing the pathways associated with genes in the strains and measuring again how the strains would respond to the IFN-γ stimulus.

## Figures and Tables

**Figure 1 biomedicines-10-02215-f001:**
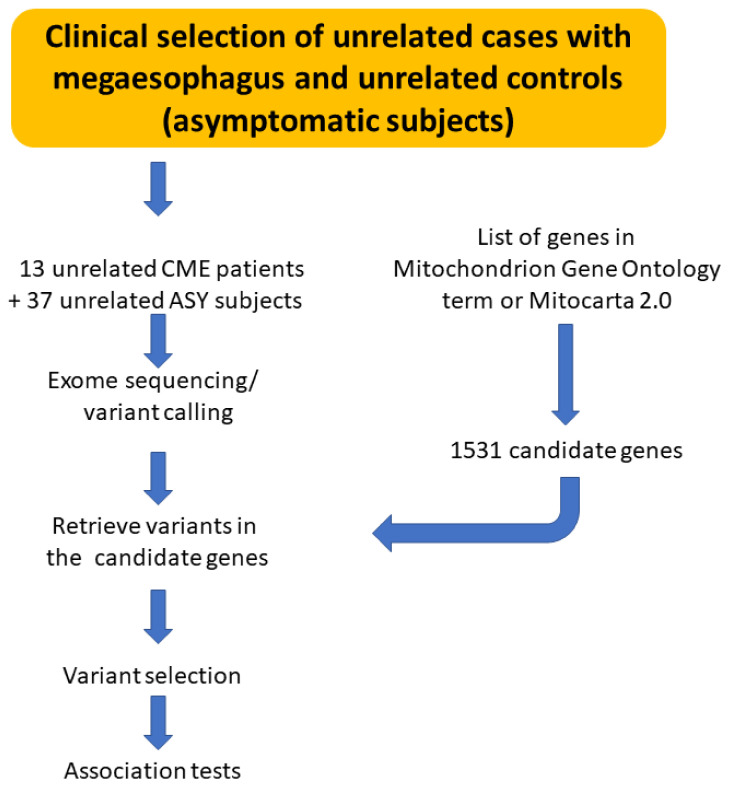
Schematic drawing describing the workflow of analysis.

**Figure 2 biomedicines-10-02215-f002:**
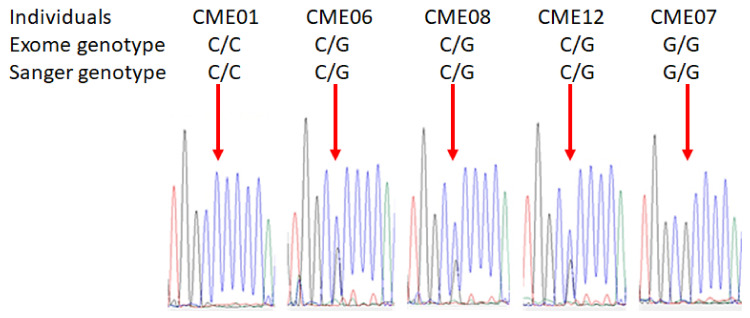
**Sanger chromatograms of patients with or without the damaging variant *MRPS18B*****.** The confirmatory sequencing was performed by Sanger sequencing. The DNA of each patient was extracted using QIAamp DNA Blood Maxi Kit (Qiagen, Germantown, WI, USA) and 25 ng of DNA was used for the amplification of MRPS18B gene. PCR quality was assessed by gel electrophoresis and all samples generated a single band corresponding to MRPS18B. Then, samples were sequenced by Sanger sequencing by Genewiz. Representative images of Sanger chromatograms of the 5 patients reveal the change in the nucleotide. The red arrow indicates the base change from cytosine to guanine.

**Figure 3 biomedicines-10-02215-f003:**
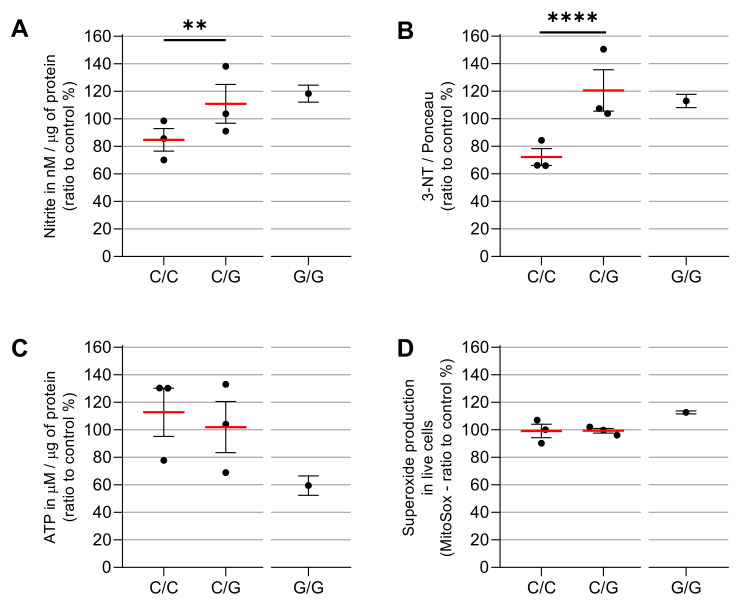
ATP production and nitro-oxidative stress after IFN-γ treatment in carriers or non-carriers of MRPS18B c.688 C > G MRPS18B p. P230A mitochondrial gene mutation. EBV-LCL were obtained from the PBMCs of the patients and infected with EBV for transformation. The EBV-LCL from the patients were stimulated in vitro with IFN-γ. We measured the production of nitrite (**A**), carried out relative quantification of nitrated proteins (**B**), ATP (**C**) and mtROS (**D**), after the stimulation with IFN-γ 25 ng/mL with MitoSOX™ Red, Griess Reagent Kit, Dot Blot, ATP Determination Kit and the anti-3-nitrotyrosine antibody. We here depict the average ratio between IFN-γ-treated vs. unstimulated cells. Each dot is a patient with ≥6 independent measurements. Experiments were repeated several times independently. As we had only one CME patient G/G, statistics were carried out only between C/C patients and C/G patients. Two-way ANOVA statistics were performed between C/C and C/G. ** = *p* < 0.01, **** = *p* < 0.0001.

**Figure 4 biomedicines-10-02215-f004:**
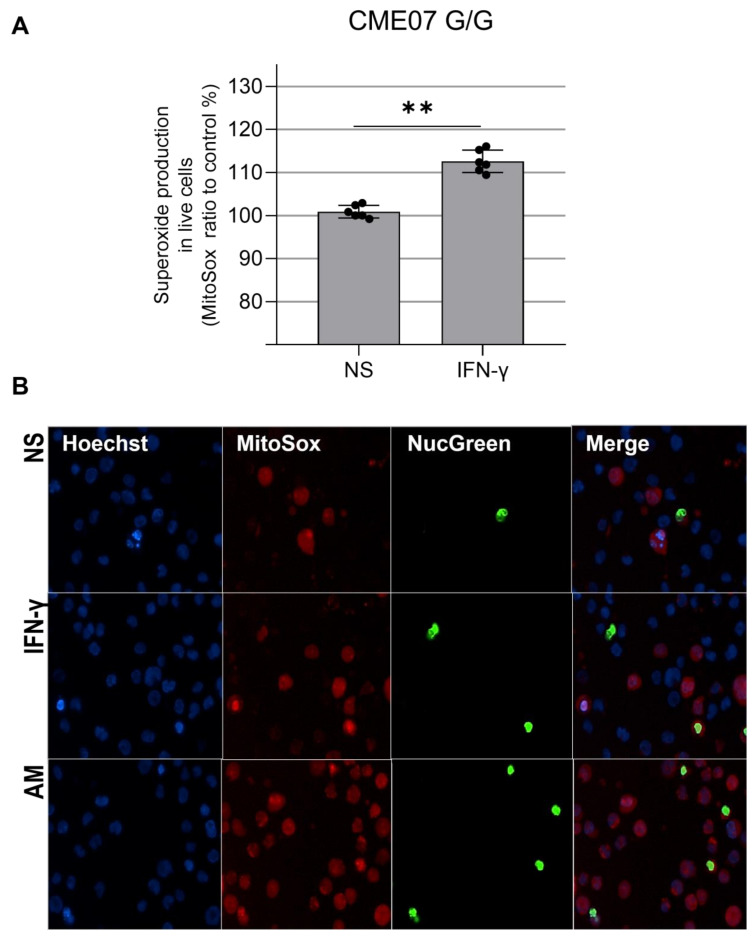
(**A**)**.** Mitochondrial ROS increase in IFN-γ-treated EBV-LCL from CME07 (homozygous MRPS18B G/G). A total of 15 × 10^4^ EBV-LCL cells were stimulated with IFN-γ 25 ng/mL for 24 h and cell nuclei were stained with Hoechst 33342 (blue), the nuclei of dead cells were stained with ReadyProbes™ NucGreen (green), also used to measure cell viability, and superoxide was measured with MitoSOX™ (red) (**B**). Positive control: 560 µM of antimycin A. Magnification 200×. NS: not stimulated. Statistical analysis performed using the Mann–Whitney test. ** = *p* < 0.01.

**Figure 5 biomedicines-10-02215-f005:**
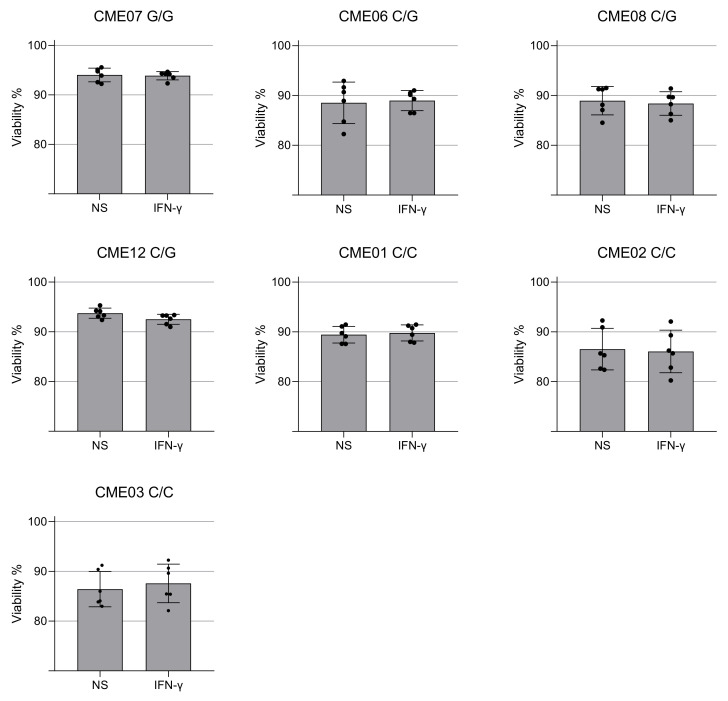
Viability of EBV-LCL cells after treatment with IFN-γ for 24 h. The cell viability of patient-specific EBV-LCL was quantified using the ReadyProbes™ NucGreen dye after 24 h of treatment with IFN-γ (25 ng/mL). Cell viability of EBV-LCL from patients with EMC with the mutation in HOM (G/G), HET (C/G) and non-carriers (C/C). Each dot is an independent measurement. Statistical analysis performed using the Mann–Whitney test.

**Table 1 biomedicines-10-02215-t001:** List of associated variants.

Gene	Ensembl Reference	Chromosome	Position	Ref Allele	Mutated Allele	rs Number	Nucleotide Change (Transcript)	Amino Acid Change (Transcript)	CADD Score
*MRPS18B*	ENSG00000204568	Chr6	30593485	C	G	rs34315095	ENST00000259873: exon7 688C > G	ENST00000259873: P230A	27.9
*FAM185A*	ENSG00000222011	Chr7	102417753	T	G	rs201667800	ENST00000409231: exon5 538T > G	ENST00000409231: Y180D	26.0

**Table 2 biomedicines-10-02215-t002:** Variants’ distribution between cases and controls.

Mutations	CME Patients			ASY Subjects		
	Heterozygote Patients	Homozygote Patients	Non-Mutated Patients	Heterozygote Subjects	Homozygote Subjects	Non-Mutated Subjects
*MRPS18B* rs34315095 C/G	4	1	8	1	0	36
*FAM185A* rs201667800 T/G	6	0	7	2	0	35

**Table 3 biomedicines-10-02215-t003:** Statistical test on mutated chromosomes.

Mutations	CME Patients			ASY Subjects		Fisher Exact Test *p*-Value
	Mutated Chromosomes	Non Mutated Chromosomes		Mutated Chromosomes	Non-Mutated Chromosomes	
*MRPS18B* rs34315095 C/G	6	20		1	73	0.001105404
*FAM185A* rs201667800 T/G	6	20		2	72	0.003611684

## Data Availability

All data are available in the manuscript.

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
