# Peer review of "Chagas Disease Megaesophagus Patients Carrying Variant MRPS18B P260A Display Nitro-Oxidative Stress and Mitochondrial Dysfunction in Response to IFN-γ Stimulus"

_biomedicines, 2022, doi:10.3390/biomedicines10092215_

Round 1
Reviewer 1 Report
In this manuscript, the authors reported an MRPS18B variant identified in Chagas disease patients and demonstrated that this MRPS18B variant could lead to increased nitrosative and oxidative stress. The findings are novel and important; however, this manuscript needs to be revised due to the smaller sample size and premature statements. A number of concerns are listed below:
1) The authors stated in the Abstract that the MRPS18B variant showed decreased ATP production, which is not supported by the data (Figure 3).
2) A Seahorse mitochondrial respiration assay on the EBV-LCL cells will be more convincing.
3) As the authors stated, Chagas disease patients develop chronic cardiac (CCC) or digestive disease, it sounds confusing when the authors hypothesize that, in mutation carriers, IFN-alpha produced in the esophageal myenteric plexus might cause nitro-oxidative stress and mitochondrial dysfunction in neurons.
Author Response
In this manuscript, the authors reported an MRPS18B variant identified in Chagas disease patients and demonstrated that this MRPS18B variant could lead to increased nitrosative and oxidative stress. The findings are novel and important; however, this manuscript needs to be revised due to the smaller sample size and premature statements. A number of concerns are listed below:
1) The authors stated in the Abstract that the MRPS18B variant showed decreased ATP production, which is not supported by the data (Figure 3).
Thank you for your comment. We actually stated in the Abstract (lines 47-48) that "the homozygous (G/G) CME patient also showed increased mitochondrial superoxide and reduced levels of ATP production", a 54% decline of ATP production not the heterozygous patients. Experiments were repeated several times independently. As we got only one CME patient G/G, statistics were done only between C/C patients and C/G patients.
Sentences were added to the revised manuscript.
2) A Seahorse mitochondrial respiration assay on the EBV-LCL cells will be more convincing.
Thank you for your comment. We agree that Seahorse mitochondrial respiration assay would provide additional data as it can discriminate mitochondrial and glycolysis-derived ATP. However, we believe that the luciferase assay is very reliable in the measurement of ATP because it is a classical chemiluminescence reaction where the luciferase enzyme depends on the cellular ATP to catalyze luciferin into oxyluciferin and generating light. Additionally, we quantified ATP by interpolation of ATP standard curve, which was performed in every single experimental plate, as our quality control strategy.
3) As the authors stated, Chagas disease patients develop chronic cardiac (CCC) or digestive disease, it sounds confusing when the authors hypothesize that, in mutation carriers, IFN-alpha produced in the esophageal myenteric plexus might cause nitro-oxidative stress and mitochondrial dysfunction in neurons.
Thank you for your comment. We agree that the hypothesis is heavily supported by the CCC studies, but this study is a starting point for chagasic megaesophagous and our hypothesis was based on two main aspects. First we have shown that IFN-gamma increases the nitro-oxidative stress in cardiomyocytes in vitro and that CCC heart tissue (a pro-inflammatory milieu, rich of IFN-gamma) displays markers of nitro-oxidative stress, such as increased nitrite and 3-nitrotyrosine and decreased mitochondrial DNA when compared with patients with dilated cardiomyopathy (https://doi.org/10.3389/fimmu.2021.755862). We hypothesized that the same phenotype could occur in megaeshopagous patients because the agent causative is the same (T. cruzi). Second, mitochondriopathies, i.e. diseases where homozygous mutations are found in the mitochondrial genes, can account for 15% of the digestive motility diseases. We found the P230A MRPS18B mutation was associated exclusively in chagasic megaesophagous patients but not in asymptomatic subjects which could gave us a clue that mitochondrial dysfunction might also be playing a role in chagasic megaesophagous progression.
Reviewer 2 Report
In this manuscript, Alcântara Silva et al. state that pathogenic mitochondrial mutations may contribute to cytokine-induced nitro-oxidative stress and mitochondrial dysfunction. Theye hypothesize that, in mutation carriers, IFN-g produced in the esophageal myenteric plexus might cause nitro-oxidative stress and mito-chondrial dysfunction in neurons, contributing to megaesophagus.
Comments:
1) The prevalence of this variant in the general population is not assessed.
2) The computational predictions of potential impacts of this variant is not addressed.
3) It would be of interest to demonstrate that MRPS18B 688C>G P230A variant is indeed pathogenic: Protein expression levels, bioenergetics…
4) Please explain how Mitosox signal is quantified and how many cells are analyzed.
5) In figure 4, results show there is no significant increase in C/G vs C/C patients, but homozygous patient CME7 (G/G) showed a 12% increase when compared
with the other patients (Figure 3D and 4A). Is it significant the 12% increase? Please specify the significance in Figure 4D.
Author Response
In this manuscript, Alcântara Silva et al. state that pathogenic mitochondrial mutations may contribute to cytokine-induced nitro-oxidative stress and mitochondrial dysfunction. They hypothesize that, in mutation carriers, IFN-g produced in the esophageal myenteric plexus might cause nitro-oxidative stress and mitochondrial dysfunction in neurons, contributing to megaesophagus.
1) The prevalence of this variant in the general population is not assessed.
Thank you for your comment.
The MRPS18B variant (688C>G, P230A) was present in 38% of CME patients (5 of the 13 CME patients) and 3% of ASY patients (1 of 37) (Table 2). One of the five CME patients carrying the mutation was a homozygous (G/G) while the other 4 CME carriers and the ASY carrier were heterozygous (C/G). So, the G allele frequency in CME is 23.1%, whereas this allelic frequency reaches 1.4% in ASY subjects.
The G variant rs34315095 has a frequency of 4.6% at the 1000 Genomes database. Based on the ALFA allele frequency project, in reference populations the alternative allele(G) frequencies were 1.8% (European reference population), 5.5% (African reference population); 5.2% (African American reference population) and 3.8% (Asian reference population). Recently, The ABraOM repository was built, containing genomic variants obtained with whole-exome and whole-genome sequencing from SABE, a census-based sample of elderly individuals from São Paulo. In both databases (exome sequencing database: 609 individuals and whole genome sequencing: 1171 individuals) the G frequency reaches 4.7%.
Sentences were added to the revised manuscript.
2) The computational predictions of potential impacts of this variant is not addressed.
Thank you for your comment. We used computationally predicted impacts (pathogenicity or damage) using 4 different algorithms: For each of these variants we look for pathogenicity data available in four databases (Shift, Polyphen2_HDIV_pred, Polyphen2_HVAR_pred and UMD predictor). We selected the variants described as pathogenic (or damaging) in at least three of these databases, lines 174-277: "For each of these variants we look for pathogenicity data available in four databases (Shift, Polyphen2_HDIV_pred, Polyphen2_HVAR_pred and UMD predictor). We selected the variants described as pathogenic (or damaging) in at least three of these databases."
The MRPS18B variant (688C>G, P230A) was characterized by a Sift score of 0.01 (damaging), a Poly-Phen score of 0.996 (damaging) and a CADD score of 28. The ENST00000259873: exon7: 688C base is well conserved in the evolution (score 8/9). The proline 230 is located in an external alpha helix.
Sentences were added to the revised manuscript.
3) It would be of interest to demonstrate that MRPS18B 688C>G P230A variant is indeed pathogenic: Protein expression levels, bioenergetics…
Thanks for the query, Indeed this would be likely next steps in this line of research. Unfortunately, it was impossible to complete these experiments in a short period of time. Regarding bioenergetics, we agree that Seahorse mitochondrial respiration assay would provide additional data as it can discriminate mitochondrial and glycolysis-derived ATP. However, we believe that the luciferase assay is very reliable in the measurement of ATP because it is a classical chemiluminescence reaction where the luciferase enzyme depends on the cellular ATP to catalyze luciferin into oxyluciferin and generating light. Additionally, we quantified ATP by interpolation of ATP standard curve, which was performed in every single experimental plate, as our quality control strategy.
We have in mind to study the effect of this variant deeper and deeper. We plan to study of this rare variant using the CRISPR/cas9 technology or on IPS cell from patient carrying or not carrying this variant. In order to study the protein expression level or the bioenergetic effect we had to increase the number of biological samples. Take in mind that only 4 CME patients were heterozygous for the variant and only one was muted homozygote.
4) Please explain how Mitosox signal is quantified and how many cells are analyzed.
Thank you for your comment. The Mitosox signal was measured using the software Columbus. First, we identified the total number of cells by using the nucleous probe Hoechst 33342. Then, dead cells were excluded by using a second probe that stains the nucleous green in dead cells. By excluding dead cells, we measured MitoSox fluorescence intensity inside live cells at approximately 580 nm (TexasRed). The contrast intensity (c) is automatically calculated by Columbus with the following function of the mean intensity in the region (a), and the mean intensity in the neighborhood of the region (b): c=(a-b)/(a+b). Approximately, 9000-10000 cells were measured per patient.
Sentences were added to the revised manuscript.
5) In figure 4, results show there is no significant increase in C/G vs C/C patients, but homozygous patient CME7 (G/G) showed a 12% increase when compared with the other patients (Figure 3D and 4A). Is it significant the 12% increase? Please specify the significance in Figure 4D.
The p value is 0.0022 Mann-Whitney test of 6 independent measurements of IFN-gamma stimulated cells compared to not-stimulated cells.
Sentence was added to the revised manuscript.
Round 2
Reviewer 1 Report
No further comments
Author Response
reviewer did not have additional comment only the associated editor
Reviewer 2 Report
The authors have adressed all my concerns
Author Response

(The authors gave the same response as above.)
